# Tumor Burden and Health-Related Quality of Life in Patients with Melanoma In-Transit Metastases

**DOI:** 10.3390/cancers15010161

**Published:** 2022-12-27

**Authors:** Hanna Wesslau, Anders Carlander, Lars Ny, Fredrik Wärnberg, Roger Olofsson Bagge, Ann-Sophie Lindqvist Bagge

**Affiliations:** 1Department of Surgery, Institute of Clinical Sciences, Sahlgrenska Academy at the University of Gothenburg, 40530 Gothenburg, Sweden; 2Department of Surgery, Sahlgrenska University Hospital, 41345 Gothenburg, Sweden; 3Wallenberg Centre for Molecular and Translational Medicine, University of Gothenburg, 40530 Gothenburg, Sweden; 4SOM Institute, University of Gothenburg, 40530 Gothenburg, Sweden; 5Department of Oncology, Sahlgrenska University Hospital, 41345 Gothenburg, Sweden; 6Department of Oncology, Institute of Clinical Sciences, Sahlgrenska Academy at the University of Gothenburg, 40530 Gothenburg, Sweden; 7Department of Psychology, Gothenburg University, 40530 Gothenburg, Sweden

**Keywords:** health-related quality of life, HRQOL, quality of life, QOL, FACT-M, melanoma, in-transit metastases

## Abstract

**Simple Summary:**

Little research has investigated how patients with melanoma in-transit metastases (ITM) experience their health-related quality of life (HRQOL). This study aimed to investigate the association between sex, age, tumor burden, and HRQOL in treatment-naïve melanoma patients with ITM. Ninety-five patients were included in this study from Sweden. The Functional Assessment of Cancer Therapy-Melanoma (FACT-M) questionnaire was used to assess HRQOL. The FACT-M consists of FACT-G (General) with the subscales PWB (Physical Well-Being), SWB (Social Well-Being), EWB (Emotional Well-Being), and FWB (Functional Well-Being) together with the melanoma-specific scales MS (Melanoma Subscale) and MSS (Melanoma Surgery Subscale). The results showed that the female sex and having more than 10 tumors were significantly associated with lower HRQOL in patients with ITM.

**Abstract:**

Background: Few studies have investigated the health-related quality of life (HRQOL) in patients with melanoma in-transit metastases (ITM). The aim was to investigate the association between tumor burden and HRQOL, including disparities pertaining to sex and age, in treatment-naïve patients with ITM. Methods: Functional Assessment of Cancer Therapy-Melanoma (FACT-M) questionnaire was used to assess HRQOL Pairwise comparisons using *t*-tests between clinical cutoffs are presented and multiple linear regression analysis showing the unique associations of gender, age, number of tumors, tumor size, presence of lymph node metastases, and tumor localization. Results: A total of 95 patients, 47% females and 53% males (median age 72 years) were included between 2012 and 2021. Women scored significantly lower on emotional well-being (*p* = 0.038) and lower on FACT-M (*p* = 0.058). Patients who had ≥10 tumors scored significantly lower on FACT-M (*p* = 0.015), emotional- and functional well-being (*p* = 0.04, *p* = 0.004, respectively), melanoma scale (*p* = 0.005), and FACT-G (*p* = 0.027). There was no significant difference in HRQOL depending on age, size of tumors, localization, or presence of lymph node metastases. Conclusion: For patients with melanoma ITMs, the female sex and higher tumor burden (i.e., number of tumors) were significantly correlated with lower HRQOL. However, these findings do not fully explain HRQOL for this patient population, and future research should consider the possibility that there are specific questions for patients with ITM where current instruments might fail to measure their discomfort to the full extent.

## 1. Introduction

Beyond cancer itself, patients with melanoma must also cope with the threat of recurrence, awareness of sun exposure, continued self-examination, and dermatological controls [1,2]. Melanoma patients who initially seek medical advice concerning pigmented lesions generally report a good health-related quality of life (HRQOL). A decline in HRQOL is observed around the time of diagnosis and the immediate post-treatment period but is thereafter slowly improved over time [1,3]. Patients having metastatic disease may be symptomatic, and often report elevated pain and fatigue combined with decreased physical and emotional functioning [1,4,5]. These questions can for example be linked to psychological distress such as depression and anxiety in relation to the often visible and/or tangible metastases, where patients can see and feel how the tumors grow and increase in number [6].

Approximately 5–10% of patients with melanoma will develop a special kind of metastasis, in-transit metastases (ITM) [7,8]. This is a type of tumor deposits within the intradermal and subcutaneous lymph vessels between the primary tumor and the nearest regional lymph node basin and typically manifests as erythematous nodules of varying sizes that may or may not be pigmented [9]. Lesions occurring within 2 cm of the primary tumor are historically classified as satellite metastases, but the current eighth version of the American Joint Committee on Cancer (AJCC) staging system groups satellite metastases and ITMs together as stage III disease, since both entities have similar tumor biology and prognosis [9,10]. ITMs can be cutaneous dermal or subcutaneous nodules, and the number of lesions can range from a single metastasis to more than a hundred. There is also a large variability in the size of the metastasis, which can range from small lesions less than 1 mm up to large and bulky lesions with several centimeters in diameter [11]. ITMs can occur in local clusters but also more extensive, e.g., covering a whole extremity [12].

Many patients with ITM are treated by simple surgical excisions, but when there are numerous lesions or short intervals between the appearance of new lesions, other treatments should be considered. There are currently several treatment options, including systemic treatment with modern immunotherapy [13], but there also exist several different loco-regional treatment options. Isolated limb perfusion (ILP) [14] and isolated limb infusion (ILI) [15,16] are two regional treatments for extremity ITM isolating the circulation of the affected limb from the rest of the body and then administrating a high dose of chemotherapy to only the affected limb. Other local treatments are intralesional injections of PV-10, a sterile solution of Rose Bengal [17,18], or talimogene laherperepvec (TVEC), a genetically modified herpes simplex virus [19,20]. 

When studying HRQOL, most studies have combined all patients with stage III disease, not taking into consideration the possible HRQOL questions arising specifically for patients with ITM [21]. These questions could for example be linked to the psychological distress regarding the often visible and/or palpable metastases, where the patients can see and feel the tumors growing and increasing in numbers [6]. These questions can for example be linked to psychological distress, such as depression and anxiety in relation to the often visible and/or palpable metastases, where patients can see and feel how the tumors grow and increase in numbers [6]. Few studies have specifically investigated HRQOL in patients with ITM [22,23,24,25,26]. The study by Yeung et al. investigated HRQOL retrospectively using FACT-M in patients after treatment with diphencycprone (DPCP) [24]. The study by McClaine et al. used a non-validated HRQOL questionnaire [25], and the study by Chin-Lenn used an HRQOL questionnaire mainly used for patients undergoing surgery for extremity tumors [26].

To our knowledge, only two studies have prospectively described HRQOL using a validated melanoma questionnaire in patients with ITM, but in small cohorts including 14 and 62 patients, respectively [22,23]. This study aims to investigate the association between tumor burden (number of tumors, largest tumor size, tumor localization, and presence of lymph node metastases) and HRQOL using the FACT-M questionnaire, including disparities pertaining to sex and age, in treatment-naïve patients with melanoma in-transit metastases.

## 2. Materials and Methods

Between October 2012 and October 2021, all patients with melanoma ITM referred to Sahlgrenska University Hospital for a first-time ILP procedure were consecutively included in this study. This center is the only site performing ILP for the Swedish population of approximately ten million people, which makes this cohort the whole patient population with ITM referred to ILP in Sweden during the study period. Patients who had received prior systemic or other locoregional treatments, except surgical excision, sentinel lymph node biopsies, or lymph node dissections, were excluded. All patients prospectively received the Functional Assessment of Cancer Therapy-Melanoma (FACT-M version 4) questionnaire before the ILP procedure. No reminders were sent out and the reasons for not responding are unfortunately not known. Bulky tumors were defined as 20 mm or larger, and a cut-off of more than 10 tumors was chosen as a marker tumor burden, these cut-offs are arbitrarily set, but are the most commonly used when defining tumor burden [16]. Data regarding patient and tumor characteristics, as well as clinical outcomes, were collected through a prospectively kept registry. Some of the patients in the present study have previously been reported by Bagge et al. [23].

The FACT-M questionnaire is a self-administered melanoma-specific HRQOL questionnaire that refers to HRQOL during the last week [5]. FACT-M is based on a definition of HRQOL as related to patients’ appraisal and satisfaction with their current level of functioning compared to perceive attainable ideals [27]. The FACT-M consists of 51 items comprising Functional Assessment of Cancer Therapy-General (FACT-G), a Melanoma Subscale (MS), and a Melanoma Surgery Scale (MSS). FACT-G is divided into four domains: Physical Well-Being (PWB), Social and family Well-Being (SWB), Emotional Well-Being (EWB), and Functional Well-Being (FWB). For all subscales, a higher score indicates a better HRQOL. Minimal important differences (MID) for FACT-M, MS, and MSS are presented in [28]. The MID indicates the smallest change in the score of a patient-reported outcome that the responder would identify as important, and would indicate a clinical change in the person’s management [29,30]. MIDs were used in this study to determine if the observed HRQOL score differs from the results of other studies. MIDs for FACT-G are presented in [31,32], and MIDs for PWB, SWB, EWB, and FWB are presented in [33].

All FACT-M scores were calculated according to the FACT-M Scoring Guidelines (version 4). Descriptive results are reported by means, standard deviations, and internal scale reliability using McDonald’s omega (ω) [34] for each subscale. Pairwise comparisons using clinical cutoffs for tumor burden and patient characteristics (sex, age) are analyzed using t-tests, and we present mean and standard deviations. Age was dichotomized according to the median age of the study population [35]. Further analyses for unique predictive effects and robustness were employed using hierarchical multiple ordinary least squares (OLS) regression analyses to model each FACT-M scale regressed on patient characteristics in step 1: age (continuous), sex (male vs. female), and jointly with tumor burden in step 2: number of tumors (continuous), and largest tumor diameter (continuous), localization (arm vs. leg), and presence of lymph node metastasis (yes vs. no). A dropout analysis was performed between the patients responding to the questionnaire or not, using Mann–Whitney U-test for continuous variables and Fisher’s exact test for proportions. Data were analyzed using Stata version 17 (StataCorp, 2021, College Station, TX, USA: StataCorp LLC) and SPSS version 27 (SPSS, Chicago, IL, USA). 

## 3. Results

### 3.1. Patient Characteristics

During 2012–2021, 165 patients with melanoma ITM were referred to Sahlgrenska University Hospital for a first-time ILP procedure. Of these, 23 patients had received prior systemic or other local treatments and were therefore excluded. All patients prospectively received the FACT-M questionnaire before the procedure, and 95 of the 142 patients (67%) returned a completed questionnaire. Of the 95 patients, 50 (53%) were male and 45 (47%) were female, with a median age of 72 years (range 40–91 years). In 83 (87%) patients the ITMs were located in the lower extremity and in 12 (13%) patients in the upper extremity (Table 1).

### 3.2. Drop-Out Analyses

A total of 142 patients were asked to fill in FACT-M, and 95 patients completed the questionnaire (67% response rate). When comparing the included patient group with the dropout group, there were no statistically significant differences in the percentage of women (64% vs. 47% *p* = 0.075), mean age (71.9 vs. 71.3 years, *p* = 0.855), ITM in the lower extremity (87% vs. 79%, *p* = 0.795), percentage of tumors over 20 mm (40% vs. 32%, *p* = 0.169), more than 10 tumors (49% vs. 34%, *p* = 0.144) or the presence of lymph node metastases (43% vs. 35%, *p* = 0.853). 

### 3.3. Descriptive Results

To get an initial picture of the results we present mean scale scores of FACT-M and the subscales in Table 2, jointly with standard deviations and internal scale reliability (McDonald’s omega). Scale scores stratified by patient characteristics (sex, below or above median age) and clinical dichotomized tumor burden groups are presented in Table 3. We specifically analysed if there was any specific threshold for number of tumors or largest tumor size, where HRQOL was affected, but no threshold could be identified, and we therefore used the commonly reported cut-offs of more or less than 10 metastases, and a tumor size of more or less than 20 mm.

### 3.4. Clinical Pairwise Comparisons

Performing pairwise comparisons using t-test showed that women (M = 135.6, 95% CI 128.1–143.1) scored lower compared to men (M = 144.0, 95% CI 138.9–149.1) on FACT-M, the difference was however not significant t(82) = 1.92, *p* = 0.058. The subscale EWB indicated a significant (t(92) = 2.10, *p* = 0.039) difference where women (M = 17.2, 95% CI 15.7–18.7) scored lower compared to men (M = 19.1, 95% CI 18.0–20.2). Yet, another significant (t(92) = 2.13, *p* = 0.036) difference was found indicating that women (M = 23.6, 95% CI 22.2–24.9) scored lower than men (M = 25.2, 95% CI 24.3–26.1) on the subscale of PWB.

Patients who had <10 tumors scored significantly (t(81) = 2.46, *p* = 0.016) higher on FACT-M compared to patients with ≥10 tumors (M = 143.8, 95% CI 139.1–148.6) vs. (M = 132.3, 95% CI 122.9–141.8). The same pattern was observed for the subscales where patients with <10 tumors scored consistently higher on EWB (M = 19.1, 95% CI 18.3–19.9 vs. M = 16.3, 95% CI 14.2–18.5, t(91) = 2.91, *p* = 0.005), followed by significant (t(91) = 2.90, *p* = 0.005) differences in FWB (M = 21.0, 95% CI 19.7–22.3) vs. M = 17.5, 95% CI 15.2–19.8) and MS (M = 55.1, 95% CI 53.4–56.8 vs. M = 50.6, 95% CI 47.6–53.6, t(92) = 2.82, *p* = 0.006), and FACT-G (M = 88.5, 95% CI 85.2–91.8 vs. M = 81.7, 95% CI 76.1–87:4, t(88) = 2.24, *p* = 0.028. Having bulky tumors (≥20 mm) was associated with a lower score in PWB (M = 23.5, 95% CI 21.7–25.3) vs. (M = 25.1, 95% CI 24.3–25.9), however not significant t(89) = 1.83, *p* = 0.070. There was no significant difference in HRQOL score depending on age, localization, or presence of lymph node metastasis.

### 3.5. Regression Analyses

Our first regression model indicates that higher FACT-M scores were significantly (R^2^ = 0.16, F(6, 71) = 3.38, *p* = 0.005) predicted in step 2 by male sex (β = 0.28, *p* = 0.010) and fewer number of metastases (β = −0.37, *p* = 0.001) when holding the other predicting variables constant. Approximately 16% of the variance in FACT-M can be explained by the predictors in our model. 

The subscale of PWB was significantly regressed (R^2^ = 0.22, F(6, 78) = 5.06, *p* < 0.001) on the predicting variables in our model demonstrating that male sex (β = 0.31, *p* = 0.002), fewer number of tumors (β = −0.36, *p* < 0.001), and smaller tumor diameter (β = −0.27, *p* = 0.009) were associated with a higher PWB score. Approximately 22% of the variance in PWB can be attributed to the predictors in the model. The second step in the regression model predicting EWB was significant (R^2^ = 0.12, F(6, 78) = 2.87, *p* = 0.014) and it was mainly male sex (β = 0.29, *p* = 0.007) and fewer number of tumors (β = −0.23, *p* = 0.029), that indicated a higher EWB score. Similarly, FWB was significantly (R^2^ = 0.10, F(6, 78) = 2.61, *p* = 0.023) regressed mainly on the number of tumors (β = −0.36, *p* = 0.001). 

The subscale of MS was significantly (R^2^ = 0.19, F(6, 79) = 4.37, *p* < 0.001) regressed on the predictors in step 2 showing that male sex (β = 0.25, *p* = 0.014), higher age (β = 0.23, *p* = 0.023), and fewer number of tumors (β = −0.40, *p* < 0.001) was related to a higher MS score. Finally, the MSS subscale was significantly regressed (R^2^ = 0.30, F(6, 79) = 6.74, *p* < 0.001) on several predictors in our model demonstrating that higher MSS scores were associated with male sex (β = 0.22, *p* = 0.026), higher age (β = 0.43, *p* < 0.001), fewer number of tumors (β = −0.24, *p* = 0.016), but was not significantly predicted by relatively smaller tumors (β = −0.18, *p* = 0.070). The regression model predicting MSS shows the highest explanatory power where approximately 30% of the variance in MSS can be explained by the predictors in our model. Full details of the regression models are presented in Appendix A. There was no evidence of multicollinearity between the predicting variables (VIF < 2 on all models). 

## 4. Discussion

This study aimed to investigate the association between tumor burden and HRQOL assessed by the FACT-M questionnaire, including differences associated with sex and age, in treatment-naïve patients with melanoma ITM.

If compared to the minimal important differences (MIDs) for FACT-M [28], the present results concerning melanoma-specific HRQOL (FACT-M, MS, and MSS) are in line with the prospective study of Jiang et al. investigating HRQOL in patients with melanoma ITM [22], with patients with ITM after they had received DPCP [24], and generally for patients stage III melanoma [21]. According to MIDs, the present ITM-treatment-naïve patients scored higher on FACT-M than the mean score of FACT-M for patients with melanoma in all stages [21]. This observed difference is likely due to the fact that the mean score for FACT-M was significantly reduced by the stage IV patients’ very low FACT-M scores. A systematic review summarizing FACT-M in patients with cutaneous melanoma in all stages indicated that FACT-M total scores were inversely correlated with the AJCC stage [21].

Compared to data in the systematic review [21], the MS-score in the present patients with ITM do not differ from patients in stage III or melanoma in all stages. These results indicate that patients with ITM do not experience lower HRQOL concerning melanoma-specific HRQOL matters than patients with stage III melanoma or melanoma in all stages. 

Compared to melanoma patients in all stages [21], the present ITM-treatment-naïve patients scored lower on MSS. Compared to data in the systematic review [21], the present ITM-treatment-naïve patients’ MSS scores are comparably low compared to all patients in stage III. The systematic review observed that the MSS score was lowest in stage III patients, probably reflecting more advanced surgical procedures in this group of patients [21].

MIDs of 5–7 points in FACT-G have been associated with important health status changes [31,32]. Based on the stated MIDs for FACT-G, the present results are in line with that of all patients with melanoma in stage III [21]. The present treatment-naïve patients with ITM scored higher on FACT-G than melanoma patients in all stages [21] which could be explained by the fact that melanoma patients in stage IV scored low on FACT-G, thus reducing the mean FACT-G score for melanoma patients in all stages. According to MIDs [31,32] the present treatment-naïve patients with ITM however scored lower on FACT-G than ITM-treatment-naïve patients in the study by Jiang et al. [22] and then the general Swedish population [36].

Based on established MIDs of 2–3 points for each of the subscales [33], the present results are in line with that of other ITM-treatment-naïve patients [22], all melanoma patients in stage III, and melanoma patients in all stages [21]. The present results however showed that treatment-naïve patients with ITM scored higher on SWB than did the general Swedish population [36]. This difference could reflect that many cancer patients experience a high rate of social support from their families, friends, and relatives [37].

In this study, treatment-naïve women with ITM scored lower on FACT-M than treatment-naïve men with ITM; where the subcategory EWB indicated the largest mean difference. A gender difference was not investigated in the ITM studies by Jiang et al. [22] nor in the study by Yeung et al. [24]. A gender difference was neither specifically studied in the systematic review investigating HRQOL in patients with melanoma [21]. The reasons why women suffering from ITM scored lower HRQOL than men suffering from ITM are still unexplained. When investigating HRQOL in the general Swedish adult population by using FACT-GP (an HRQOL-generic instrument equivalent to FACT-G) the results showed that women in the general Swedish population scored significantly lower on FACT-GP than men [36]. A similar gender difference was also observed when assessing HRQOL by EORTC-QLQ-C30 in a German general population [38] and in a Swedish general population [36]. A suggested explanation for the gender differences is that men can potentially be less inclined to admit an impaired QOL as compared to women [39].

There was no significant difference in HRQOL score depending on age. Age was not an investigated factor in relation to HRQOL in the comparable ITM studies by Jiang et al. [22] and Yeung et al. [24]. It should be noted that the mean age in the present study of patients with ITM is high (72 years), as it is in the comparable ITM studies by Jiang et al. (70 years) and by Yeung et al. (78 years). The present FACT-G score is comparable to the FACT-GP score of the general Swedish population in the age group of 65+, indicating that treatment-naïve patients with ITM do not experience lower non-melanoma specific HRQOL than the age-matched general population [36].

The present results showed that patients with ITM who had ≥10 tumors scored significantly lower in FACT-M compared to patients with <10 tumors, where the subcategories: EWB followed by significant differences in FWB, MS, and, FACT-G, indicated the largest mean differences. The present results further showed that patients with ITM having bulky tumors scored lower in PWB, although not significant and this difference did not affect the overall FACT-M score. The present results, however, showed no significant differences in HRQOL score depending on the localization of metastasis or the presence of lymph node metastasis. Tumor characteristics were not related to HRQOL in the comparable ITM studies by Jiang et al. [22] and Yeung et al. [24].

The present result concerning the gender difference implies that female patients with ITM scored lower HRQOL than men. This gender difference concerning HRQOL is observed in some other cancer populations [40] as well as in the general population [36,38,41]. Many reasons have been put forward to explain why women rate their HRQOL as worse than men, including built-in societal inequalities between the sexes. However, the result may also reflect that men might be less inclined to admit an impaired QOL as compared to women [39]. It is thus of clinical relevance to also note men’s well-being to meet their possibly unmet care needs, even if they do not verbally indicate a deterioration in well-being. In fact, there are studies showing that men have delayed help-seeking behavior compared to women [42].

This patient cohort consists of the entire patient population of treatment-naïve patients with melanoma ITMs in Sweden. Thus, we could not affect the sample size. A priori sample size calculation for a two-tailed Student’s t distribution assuming a small (Cohen’s d) effect size of 0.2, power (1−β) of 0.08, error rate (α) of 5% results in a recommended sample size of approximately 800 individuals, and based on normative data for a large cancer sample the mean difference between for example men and women on FACT-G would require a total sample size of approximately 1000 patients [43]. However, we observe larger mean group differences in the current data between men and women on the total FACT-M corresponding to a medium (Cohen’s d) sized effect of 0.41, which render a required sample size of approximately 200 individuals. This indicates that the current study may be underpowered. Still, for regression assuming a medium (f^2^) effect size of 0.15, power (1−β) of 0.08, error rate (α) of 5%, and six predictors the current sample of 95 patients is close to the required minimum sample size of 97 individuals. 

The present results further indicate that increased tumor burden (i.e., number of tumors) is associated with reported lower HRQOL in patients with ITM melanoma. Patients with ITM have previously been reported to experience increased anxiety and pain associated with their increasing tumor burden [6]. Anxiety and depression have been observed to be strongly related to unmet care needs in melanoma patients. In fact, a quarter of melanoma patients with stage I-III report unmet care needs partly related to anxiety and/or depression [44]. Unmet care needs in cancer patients have been suggested to significantly affect patients’ HRQOL [44]. Since unmet care needs, anxiety and/or depression, and reported lower HRQOL seems to be interrelated, it is of clinical relevance to identify the individuals in this cancer population who need extra care and treatment. It is also important to alert healthcare professionals that this cancer population is at increased risk of certain complications due to their particular form of cancer, which, if not addressed properly, can lead to unmet care needs.

### Limitations/Strengths

The strength of the present study is that it is a consecutive population-based cohort of patients with melanoma ITM that is referred from all over Sweden. The limitation is that the study is based on patients with melanoma ITM that had been referred specifically to undergo treatment with ILP, thus might not reflect the entire group of patients with melanoma ITM. E.g., these patients have ITM localized to the limbs only, and might potentially suffer from a different disease burden than other patients. These patients have also actively been referred to a national center, which might add further bias to the population. The study had a response rate of 70%, and analyses showed that there was no significant difference between the group of patients participating in this study compared to the drop-out group.

In hindsight, another limitation is that the study did not include measures of anxiety and depression. These had been important variables to measure as these mental difficulties have been reported in patients with ITM [6]. The FACT-M does not contain subscales that specifically examine anxiety, depression, and pain. It would be possible that some individual items in FACT-M may correlate to clinically validated instruments that specifically examine anxiety, depression and pain, though we are not aware of such studies.

## 5. Conclusions

We conclude that for patients with melanoma ITM, lower HRQOL is associated with female gender and higher tumor burden (i.e., number of tumors). Men might be less inclined to admit an impaired QOL compared to women, making it clinically relevant to note men’s well-being even if they do not verbally indicate a deterioration in well-being. The present result further indicates that increased tumor burden (i.e., number of tumors) is associated with a reported lower HRQOL in patients with ITM. It is important to alert healthcare professionals that this cancer population is at increased risk of certain complications due to their particular form of cancer, which, if not addressed properly, can lead to unmet care needs. Future research should aim at taking into consideration the possibility that there are specific HRQOL questions arising specifically in patients with ITM, with often visibly and palpably growing metastases that tend to get ulcerous and bulky are of great concern, and where current HRQOL instruments might not measure this discomfort to the full extent. 

## Figures and Tables

**Table 1 cancers-15-00161-t001:** Patient characteristics and tumor burden.

		Sex	
		Female (n)	Male (n)
Age	≤72 yrs	22	27
	>72 yrs	23	23
Lymph node metastases	No	27	28
	Yes Missing	15 3	18 4
Tumor localization	Leg	41	42
	Arm Missing	4 0	8 0
Numbers of tumors	<10	32	30
	≥10 Missing	13 0	19 1
Tumor size	<20 mm	33	29
	≥20 mm Missing	12 0	18 3

**Table 2 cancers-15-00161-t002:** Mean scale scores (M), standard deviations (SD), and reliability (ω) of the total FACT-M scale and the subscales FACT-General (FACT-G), Physical well-being (PWB), Social well-being (SWB), Emotional well-being (EWB), Functional well-being (FWB), Melanoma scale (MS), and Melanoma surgical scale (MSS).

Scales	M	SD	ω
FACT-M	140.1	20.4	0.91
FACT-G	86.2	13.9	0.95
PWB	24.4	3.8	0.81
SWB	23.8	4.6	0.90
EWB	18.1	4.5	0.83
FWB	19.8	5.8	0.90
MS	53.5	7.6	0.81
MSS	23.6	6.7	0.87

**Table 3 cancers-15-00161-t003:** Mean and standard deviation. FACT-General (FACT-G), Physical well-being (PWB), Social well-being (SWB), Emotional well-being (EWB), Functional well-being (FWB), Melanoma scale (MS), and Melanoma surgical scale (MSS).

		PWB	SWB	EWB	FWB	MS	MSS	FACT-G	FACT-M
Sex	Female	23.6 ± 4.39 *	23.8 ± 4.47	17.2 ± 4.96 *	19.1 ± 5.75	52.4 ± 8.32	22.4 ± 7.04	84.0 ± 15.07	135.6 ± 23.15
	Male	25.2 ± 3.12	23.8 ± 4.84	19.1 ± 3.83	20.5 ± 5.82	54.6 ± 6.75	24.5 ± 6.36	88.3 ± 12.40	144.0 ± 16.97
Age	<72	24.3 ± 4.60	23.8 ± 4.94	17.7 ± 5.04	20.5 ± 6.77	52.5 ± 9.06	21.6 ± 7.36 **	86.3 ± 16.58	138.5 ± 24.61
	>72	24.6 ± 2.82	23.7 ± 4.33	18.6 ± 3.79	19.1 ± 4.48	54.7 ± 5.45	25.9 ± 5.05	86.1 ± 9.98	142.0 ± 13.77
Number of tumors	<10	24.9 ± 3.59	23.7 ± 4.88	19.1 ± 3.10	21.0 ± 5.18	55.1 ± 6.75	24.1 ± 5.81	88.5 ± 12.58	143.8 ± 17.77
	≥10	23.6 ± 4.28	23.8 ± 4.24	16.3 ± 6.05 **	17.5 ± 6.34 **	50.6 ± 8.33 **	22.9 ± 7.93	81.7 ± 15.43 *	132.3 ± 23.82 *
Tumor diameter	<20 mm	25.1 ± 3.09	23.8 ± 4.80	18.1 ± 4.83	20.1 ± 5.94	54.3 ± 6.84	24.4 ± 6.14	87.2 ± 13.87	141.8 ± 19.92
	≥20 mm	23.5 ± 4.88	23.6 ± 4.53	18.8 ± 3.27	19.5 ± 5.55	52.7 ± 8.87	22.2 ± 7.11	85.2 ± 14.08	138.0 ± 21.26
Tumor localization	Leg	24.3 ± 4.02	23.8 ± 4.79	17.8 ± 4.63	19.7 ± 5.96	53.1 ± 7.80	23.2 ± 7.02	85.6 ± 14.30	138.9 ± 21.11
	Arm	25.6 ± 1.98	23.7 ± 3.59	20.5 ± 2.16	21.0 ± 4.40	56.4 ± 5.12	26.1 ± 3.55	90.5± 9.43	147.7 ± 13.06
Presence of lymph node metastases	No	24.5 ± 4.05	23.5 ± 5.10	18.9 ± 4.12	20.0 ± 6.01	53.9 ± 7.62	24.8 ± 6.60	86.7± 14.48	140.6 ± 20.90
	Yes	24.9 ± 3.49	24.3 ± 4.10	17.1 ± 5.02	19.9 ± 5.81	53.4 ± 7.78	23.7 ± 6.88	86.5 ± 13.39	140.7 ± 20.04

* *p* < 0.05, ** *p* < 0.01.

## Data Availability

Anonymized data can be shared upon reasonable request.

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
