# Peer review of "Tumor Burden and Health-Related Quality of Life in Patients with Melanoma In-Transit Metastases"

_cancers, 2022, doi:10.3390/cancers15010161_

Round 1

Reviewer 1 Report

I thank the academic editor for giving me the pleasure of reviewing this interesting paper in which the authors conduct a study on 95 patients diagnosed with in-transit metastasis (ITM) from Malignant Melanoma and that were naïve for treatment. In this way, the authors administer a validated questionnaire called FACT-M to detect the health-related quality of life (HRQOL) in patients suffering of ITM.

I think that the paper would be mostly suitable for a psychological journal, but, because the paper is scientifically valid, could be published also in the prestigious journal “Cancers”.

Following my suggestions to improve the manuscript:

1.       In the Introduction section is necessary to add one or two sentences about the histopathological diagnosis of Malignant Melanoma and/or in-transit metastasis. I suggest following papers:

1.       Cazzato G, Colagrande A, Maruccia M, Nacchiero E, Lupo C, Casatta N, Ingravallo G, Maiorano E, Marzullo A, Giudice G, Resta L. Polymorphic Malignant Melanoma (PMM) of the Left Helix: Case Report with Clinical-Pathological Correlations. Diagnostics (Basel). 2022 Nov 6;12(11):2713. doi: 10.3390/diagnostics12112713. PMID: 36359557.

2.       Friedman RJ, Heilman ER. The pathology of malignant melanoma. Dermatol Clin. 2002 Oct;20(4):659-76. doi: 10.1016/s0733-8635(02)00041-4. PMID: 12380053.

3.       Check the entire manuscript for some typos and English language.

Author Response

Comment: In the Introduction section is necessary to add one or two sentences about the histopathological diagnosis of Malignant Melanoma and/or in-transit metastasis. I suggest following papers:  Cazzato G et al. Polymorphic Malignant Melanoma (PMM) of the Left Helix: Case Report with Clinical-Pathological Correlations. Diagnostics (Basel). 2022 Nov 6;12(11):2713. 2.  Friedman RJ, Heilman ER. The pathology of malignant melanoma. Dermatol Clin. 2002 Oct;20(4):659-76.

Reply: Thank you for the comment, we have now added more information about histopathological diagnostics of in-transit metastasis in the second paragraph of the Introduction, including new references.

Comment: Check the entire manuscript for some typos and English language.

Reply: We have checked the manuscript and improved the English language.

Reviewer 2 Report

From the discussions it appears that this study confirms similar observations and gives an estimate on the problem in the respective country.

The article is the expression of a well-done study that achieved its goal.

Author Response

From the discussions it appears that this study confirms similar observations and gives an estimate on the problem in the respective country. This article is the expression of a well-done study that achieved its goal.

Reply: Thank you for your positive feedback and for taking the time to revise the paper.

Reviewer 3 Report

The topic of the paper is interesting. However, the writing style needs work. Much of the paper is dry and boring with limited justification, integration, or implications provided.

Abstract – The Conclusion should include implications.

Intro

1.       Provide more info about ITM and ILP to better justify the focus of the paper.

2.       Suggest mentioning the visible and palpable metastases earlier and perhaps in more detail.

3.       Briefly mention the findings of the prior HRQOL studies.

Methods

1.       Describe how the cut-offs were selected. The use of the terms and the purpose of clinical, cut-off, and clinical cut-off are unclear.

2.       Write out MID upon first occurrence. Say how these will be used.

3.       Is the sample size large enough to find differences in all the comparisons?

Results

1.       The Results are mostly a list. There could be better summaries, integration, transitions, etc.

2.       Please write out the scale abbreviations or use a brief name for easier readability.

3.       Would it be worth including some of the excluded patients if they don’t differ too much from the current sample?

4.       It’s unclear what the “sensitivity” analyses were.

5.       Drop-out analyses are often reported at the beginning of the Results section and more briefly.

Table 3 – Suggest removing the CIs and noting significant differences instead.

Figure 1 – Not sure this is the optimal way to show these findings.

Discussion

1.       The Discussion is dry and boring.

2.       Unless the Clinical Implications (and Conclusions) are required to be separate, integrating them with the prior sections would help make them less dry.

3.       The explanation for the gender difference is one possibility. However, there are several other likely ones such as that women’s HRQOL is actually worse, perhaps due to sexism, etc.

4.       Did any of the scales address anxiety, depression, or pain? Regardless, these topics should also be mentioned in the Intro rather than first in the Discussion.

5.       More context could be provided by comparing the scores here to patients with other cancers or skin conditions.

Author Response

Comment: The topic of the paper is interesting. However, the writing style needs work. Much of the paper is dry and boring with limited justification, integration, or implications provided.              
Reply: We apologize if the manuscript seemed dry and boring. We have tried to improve the text based on your comments, and we have improved the English language

Comment: Abstract – The Conclusion should include implications.

Reply: The potential implications of this study have now been added.

Comment: (Intro. 1). Provide more info about ITM and ILP to better justify the focus of the paper

Reply: Thank you very much for this comment. We have now provided more information on ITM and briefly mentioned the different treatments available for ITM melanoma (2nd and 3rd paragraph of the Introduction). However, we still only briefly cover ILP since this was not the aim of this study (all patients filled in the questionnaire before treatment).

Comment: (Intro. 2). Suggest mentioning the visible and palpable metastases earlier and perhaps in more detail.

Reply: We agree with your remark, and the information has now been added in the 1st paragraph of the Introduction.

Comment: (Intro. 3). Briefly mention the findings of the prior HRQOL studies.

Reply: We briefly mentioned relevant published HRQOL studies in the 4th and 5ft paragraph of the Introduction, and we now discuss these more thoroughly in the Discussion.

Comment: (Methods. 1). Describe how the cut-offs were selected. The use of the terms and the purpose of clinical, cut-off, and clinical cut-off are unclear.

Reply: The cut-offs used are arbitrarily used for historic reasons, where most other reports use the same cut-offs. We have more in detailed discussed this more in the 1st paragraph of the Method section and included a reference. However, we have also analyzed the data, and we could not identify any obvious threshold for either the number of tumors or the largest tumor size, and we now mention this shortly in the Results section (paragraph 3.3).

Comment: (Methods. 2). Write out MID upon first occurrence. Say how these will be used.

Reply: Thank you for the remark.  We have now added an explanation of MID in the 2nd paragraph of the Method section and MID is written out when it first appears.

Comment: (Methods. 3). Is the sample size large enough to find differences in all the comparisons?

Reply: Thank you very much for your comments. We have now added a part in the Discussion where we thoroughly discuss this issue, including power calculations for the analyses. With pairwise comparisons, the sample size is perhaps underpowered but was close to the minimum sample size of 97 patients.

Comment: (Results. 1). The Results are mostly a list. There could be better summaries, integration, transitions, etc.

Reply: We have discussed this issue of structuring by subheadings, and we have now better introduced the reader to the different subheadings. Please see the updated Results section.

Comment: (Results. 2). Please write out the scale abbreviations or use a brief name for easier readability.

Reply: We have now added the whole scale abbreviations to clarify for the reader.

Comment: (Results. 3). Would it be worth including some of the excluded patients if they don’t differ too much from the current sample?

Reply: If patients in the study for unknown reasons choose not to participate and fill in the questionnaire, we were not able to include them in this study. Other patients that were excluded had undergone prior systemic treatments, making them not treatment-naïve and therefore not eligible for the aim of this study.

Comment: (Results. 4). It’s unclear what the “sensitivity” analyses were.

Reply: The concept of sensitivity analyses can sometimes be misleading. After discussion, we have erased the term and only mention the regression analyses where we keep the variables for robustness testing.

Comment: (Results. 5). Drop-out analyses are often reported at the beginning of the Results section and more briefly.

Reply: We agree that the drop-out analyses should come earlier in the result section, and we have now corrected this. We have also shortened the drop-out analyses section.

Comment: (Results. Table 3). Suggest removing the CIs and noting significant differences instead.

Reply: Thank you for this comment. We do agree that the table is somewhat difficult to overview. We have now removed the CIs from table 3 and instead noted the significant differences with asterisks to create a more readable table.

Comment: (Results. Figure 1). Not sure this is the optimal way to show these findings.

Reply: We do agree that this figure is more appropriate in the context of a seminar or poster session. We have therefore omitted it from the manuscript.

Comment: (Discussion. 1). The Discussion is dry and boring.

Reply: We have tried to address your comment by rewriting the discussion with more reasoning and with less repetition.

Comment: (Discussion. 2). Unless the Clinical Implications (and Conclusions) are required to be separate, integrating them with the prior sections would help make them less dry.

Reply: We have shortened the clinical implications and integrated them with the prior discussion section to make it easier readability.

Comment: (Discussion. 3). The explanation for the gender difference is one possibility. However, there are several other likely ones such as that women’s HRQOL is actually worse, perhaps due to sexism, etc.

Reply: The reviewer is correct that there may be other explanations for the observed gender difference than what was stated in the manuscript. We have now developed the discussion around gender differences.

Comment: (Discussion. 4). Did any of the scales address anxiety, depression, or pain? Regardless, these topics should also be mentioned in the Intro rather than first in the Discussion.

Reply: To introduce the concepts of anxiety, depression, and pain in patients with ITM earlier, we have now revised the sentence in the introduction. We have further addressed the topic of anxiety and depression in relation to the used HRQOL instrument FACT-M in the limitations in the Discussion.

Comment: (Discussion. 5). More context could be provided by comparing the scores here to patients with other cancers or skin conditions.

Reply: Regarding FACT-M, which is a diagnosis-specific HRQOL questionnaire, we have compared all relevant HRQOL papers. According to a published systematic review referred to in this study, it is difficult to compare different HRQOL studies to each other if they have not used the same HRQOL instrument since there is not a single definition of HRQOL. It, therefore, makes it somewhat problematic to compare our data with data from other studies based on patients with other diagnoses where these studies have used other diagnosis-specific HRQOL questionnaires than the FACT-M used in this study.